# Clinical Pharmacists’ Knowledge of and Attitudes toward Pharmacogenomic Testing in China

**DOI:** 10.3390/jpm12081348

**Published:** 2022-08-21

**Authors:** Xiaoyan Nie, Tong Jia, Xiaowen Hu, Sicong Li, Xinyi Zhang, Caiying Wu, Yuqing Zhang, Jing Chen, Luwen Shi, Christine Y. Lu

**Affiliations:** 1Department of Pharmacy Administration and Clinical Pharmacy, School of Pharmaceutical Sciences, Peking University, Beijing 100191, China; 2International Research Center for Medicinal Administration, Peking University, Beijing 100191, China; 3Department of Population Medicine, Harvard Medical School and Harvard Pilgrim Health Care Institute, Boston, MA 02215, USA

**Keywords:** pharmacogenomic testing, clinical pharmacist, knowledge, attitude, China

## Abstract

(1) Background: Uptake of pharmacogenomic testing in routine clinical practices is currently slow in China. Pharmacists might play an important role in leveraging care through applying pharmacogenomics, therefore, it is important to better understand clinical pharmacists’ knowledge of and attitudes toward pharmacogenomic testing, which has not been well-studied. (2) Methods: A self-administered survey was developed based on previous knowledge of pharmacogenomic testing and its uptake in China. Participants were recruited through the Committee of Pharmaceutical Affairs Management under the Chinese Hospital Association. (3) Results: A total of 1005 clinical pharmacists completed the questionnaire, among whom 996 (99.10%) had heard of pharmacogenomic testing before participation. More than half of respondents (60.0%, *n* = 597) rated their knowledge of pharmacogenomic testing as “average”, while 25% rated it “good” or “excellent”. “Guidelines, consensus and treatment paths for disease diagnosis and treatment” (78.7%) were the most preferred sources of information about pharmacogenomic testing. Most respondents (77.0%) believed that pharmacogenomics could “help to improve efficacy and reduce the incidence of adverse reactions”. Our participants also believed that patients would benefit most from pharmacogenomic testing through better prediction of individual drug responses and thus informed treatment decisions. The top challenge for the uptake of pharmacogenomic testing was its high cost or lack of insurance coverage (76.7%). (4) Conclusions: Most Chinese clinical pharmacists who participated in our study had a positive attitude toward pharmacogenomic testing, while the knowledge of pharmacogenomic testing was generally self-assessed as average.

## 1. Introduction

Pharmacogenomics (PGx) research and its implications in clinical practices have increased in recent years as healthcare moves towards precision medicine [1]. PGx testing is the use of genetic tests to suggest the optimal pharmaceutical therapy for individual patients [2], which has the potential to reduce adverse drug responses and improve drug efficacy. To date, the US Food and Drug Administration (FDA) has approved more than 400 drug labels with information on genomic biomarkers and their relationship with drug exposure, dosage, risk of adverse effects, and clinical response variability [3]. Moreover, the Dutch Pharmacogenetics Working Group (DPWG), the Clinical Pharmacogenetics Implementation Consortium (CPIC), the Canadian Pharmacogenomics Network for Drug Safety (CPNDS), and the French National Network (Réseau) of Pharmacogenetics (RNPGx) have provided evidence-based clinical recommendations for PGx practice [4]. 

Increasing uptake of PGx testing has been most notably witnessed in the fields of oncology and cardiovascular diseases and gradually in the fields of psychiatry [5,6], pain relief [7], rheumatic immunology [8], and neurology [9,10]. However, in China, PGx testing has yet to be fully incorporated into routine clinical practice.

Clinical pharmacists have unique advantages in managing drug therapies. Previous studies suggested that clinical pharmacists are well-positioned to promote uptake of PGx and leverage care. The American Society of Health-System Pharmacists (ASHP) has also called for “all pharmacists have a responsibility to take a prominent role in the rational, ethical use and clinical application of pharmacogenomic” [11]. However, pharmacists in high-income countries reported limited knowledge of and many concerns about PGx; high costs, limited insurance reimbursement, lack of domestic clinical guidelines, lack of trust in the vendor of PGx results, and fear of discrimination and misinterpretation of test results were among factors commonly recognized to have hindered PGx’s uptake in clinical practices [12]. In 2019, Guo et al. [13] designed a questionnaire to obtain a comprehensive understanding of PGx by physicians, pharmacists, and researchers. However, given its small sample size of pharmacists and the fact that the respondents mainly resided in south-central and south-western regions, such as Hunan Province, and did not distinguish between types of pharmacists, there is a lack of research on the clinical pharmacists’ knowledge of and attitudes toward PGx in China. Thus, this study aimed to explore clinical pharmacists’ attitudes toward and knowledge of PGx testing in China and their views on the main barriers that had hindered uptake of PGx testing.

## 2. Materials and Methods

### 2.1. Survey Development

Based on the current use of PGx testing in China and similar studies [14,15,16,17,18,19,20] in other countries, we developed a questionnaire inquiring clinical pharmacists’ knowledge of, attitudes toward, and experience with PGx testing. The survey contains 38 questions and takes an estimated 10–15 min to complete. The questionnaire has been mentioned in our previously published study [21]. In this paper, we report findings from the first and third sections (24 questions) of the survey, focusing on the attitude and knowledge of pharmacists. The 14 questions in the survey’s second section pertained to clinical pharmacists’ involvement in PGx testing, which will be presented in another article. If the respondent selected “I volunteer to participate in the study”, all questions should be answered. (More details on the methods can be found in Appendix B). 

### 2.2. Sampling Methods and Data Collection

The sample was recruited through the Committee of Pharmaceutical Affairs Management under the Chinese Hospital Association, the organizer of a training platform for clinical pharmacists. The participants used the “Wenjuanxing” platform (www.wjx.cn (accessed on 10 October 2021)) [22] to complete the electronic questionnaire, which was sent to the working group of the training platform according to the distribution of provinces. The questionnaires answered by clinical pharmacists who met the two inclusion criteria (Appendix B) were included in this study.

The questionnaire was open for 30 days until 12 November 2021. The study was approved by the Institutional Review Boards at Peking University, Beijing, China (IRB 2021100). Respondents who completed the survey received 10 yuan (~US$ 1.48) as compensation for their time.

### 2.3. Data Analysis

For each question, response frequencies and percentages were described. Factors that might be associated with the knowledge of and attitudes toward PGx testing were analyzed by multivariate logistic regression analysis. Estimates were weighted with a 95% confidence interval provided, where applicable. All statistical analyses were performed using STATA 15.1.

## 3. Results

### 3.1. Survey Completion Rates and Inclusion

Our electronic questionnaire had 1560 clicks, however, 555 were excluded as they were from pharmacists who did not work at Chinese tertiary hospitals, or were not clinical pharmacists; questionnaires that were not completed were also excluded. In the end, the survey completion rate was 64.4%, resulting in a total of 1005 clinical pharmacist responses from 31 provinces and autonomous regions were included in the study.

### 3.2. Respondent Characteristics

Among the total of 1005 clinical pharmacists, 996 (99.1%) had heard of PGx testing before participation (who hereafter would be referred to as the pre-knowns), and nine (0.9 %) had not. Characteristics of study respondents are shown in Table 1 and Appendix A. No significant differences were observed in any baseline characteristics between the total study sample and the pre-knowns. The following results are based on pre-knowns.

Considering that respondents’ willingness to adopt new technologies might impact their attitudes toward PGx testing, a relevant question was included in the questionnaire. Most respondents (80.8%, *n* = 812) selected “aware of the need to change and very comfortable adopting new technologies and adopt new technologies before the average person, but need to see evidence of success before adopting”. Meanwhile, 119 (11.8%) and 74 (7.4%) respondents chose “want to be the first person to try an innovation” and “skeptical of change, only adopt an innovation after it has been tried by the majority”, respectively.

### 3.3. Knowledge of PGx Testing

Many respondents (59.9%, *n* = 597) rated their knowledge of PGx testing as “average”, while 24.6% rated “good” or “excellent” (Figure 1).

The multivariate analysis for clinical pharmacists’ knowledge with PGx testing showed that clinical pharmacists with the following baseline characteristics showed better self-evaluated knowledge of PGx testing: male (*p* < 0.001), with a master’s degree (*p* < 0.001) or doctoral degree (*p* < 0.001), were associate chief or chief clinical pharmacists (*p = 0.01*) or with a middle-level title (*p* = 0.02), in provinces with high GDP rank (*p* < 0.001), or hold a positive attitudes toward new technologies (*p* < 0.001).

Clinical pharmacists’ knowledge about different PGx-related guidelines, monographs, and databases varied. In general, their knowledge of domestic guidelines or expert consensus was better, with half of the respondents (50.9%, *n* = 507) rating themselves “good” or “excellent” on this aspect of their PGx knowledge. Clinical pharmacists’ knowledge about international guidelines and databases was relatively low, and only 17.5% (*n* = 174), 25.7% (*n* = 256), 23.6% (*n* = 235) rated themselves as having “good” or “excellent” knowledge about ClinGen, PharmGKB, and CPIC, respectively (Figure 2).

We analyzed the interactions between the level of knowledge about PGx and of the corresponding guidelines, monographs, and databases. For instance, 53.0% of the respondents who had a good or excellent grasp of the PGx testing also rated themselves as having good or excellent knowledge about the CPIC guidelines, while 88.3% of the respondents who had a lower-than-average knowledge of PGx testing also said that their knowledge about the guidelines was below average. Not surprisingly, respondents who had a higher level of knowledge about PGx testing also had a higher level of knowledge about the corresponding PGx-related guidelines, monographs, and databases (*p* < 0.001).

The majority (78.7%) of respondents had accessed information about PGx through “Guidelines, consensus and, clinical pathways for disease diagnosis and treatment”, followed by “Academic conference” (64.1%), and “Professional PGx-related guidelines, monographs, or databases” (49.7%).

### 3.4. Attitudes towards PGx Testing

Respondents generally acknowledged the significance of PGx testing. Most respondents (77.2%) agreed that PGx testing could “help to improve efficacy and reduce the incidence of adverse reactions from drug therapy”. For all questions, more than 50.0% of respondents agreed with the description presented (Figure 3).

Respondents who were positive toward new technologies were also more positive toward the significance of PGx testing in clinical practices than those who were negative toward novel technologies (*p* < 0.001). After excluding the influence of the significant factor “attitude towards new technologies” in the multivariate analysis, respondents with “excellent” knowledge of PGx testing were significantly more positive towards PGx testing’s role and clinical impact.

When being asked about the value of PGx in different therapeutic areas, the use of PGx testing in “Targeted Oncology Therapy” was most highly recognized, with more than half of respondents rating the value of PGx as 80–100 in this therapeutic area (Table 2). The association between respondents’ clinical practice specialties and their scoring of the value of PGx testing in different areas shows that clinical pharmacists in oncology departments tended to score oncology therapeutic areas higher than therapeutic areas in which they did not specialize (Table 3).

Half of respondents (50.9%, *n* = 507) agreed that patients would benefit most from PGx testing if results were used to help predict poor response to a medication, followed by those who used test results to help predict possible side effects (23.1%, *n* = 230) and those who used test results to guide the determination of the initial dose (13.7%, 136). However, there were four respondents (0.4%) who thought PGx testing would not benefit patients at all.

### 3.5. Barriers of Developing PGx Testing

Our participants suggested that the top three challenges for increasing uptake of PGx testing were its high cost or lack of insurance coverage (76.7%, *n* = 764), the lack of trained professionals (65.9%, *n* = 656), and lack of relevant knowledge (53.7%, *n* = 535). Other challenges perceived by many of the participants included lack of testing devices (46.6%, *n* = 464) and shortage of recommendations by guidelines or consensus (42.2%, *n* = 420).

### 3.6. Others

#### 3.6.1. Responsibilities of Pharmacists in PGx Testing

“Make recommendations to physicians on drug selection, dosage, and monitoring based on the results of PGx testing” (94.8%, *n* = 944) and “Interpret PGx test results to physicians and patients” (87.5%, *n* = 871) were the two most important responsibilities identified by the pre-knowns. There were also four clinical pharmacists who suggested that other responsibilities of pharmacists in PGx testing might include “Conduct related research” and “Generate clinical evidence of PGx testing”.

#### 3.6.2. Willingness to Participate in PGx Trainings

Among all respondents, 906 (90.2%) suggested that they were willing to participate in PGx training if provided, while 16 (1.6%) showed no interest. Favored training modes were “Short-term intensive training” (57.3%), “Online video courses” (56.2%), “Expert lectures” (52.3%), and “Academic conferences” (52.0%).

## 4. Discussion

To assess clinical pharmacists’ knowledge of and attitudes toward PGx testing, we conducted this survey among clinical pharmacists practicing at tertiary hospitals across mainland China. We found that most Chinese clinical pharmacists who participated in our survey had a relatively good understanding of and positive attitudes toward PGx testing compared with results presented by similar studies in Saudi Arabia [23] (knowledge score for all participants was 2.4 out of 5.0), Japan [24] (only 12.5% respondents showed a good understanding of PGx testing), Thailand [25], Netherlands [26], and Australia [27,28] (6–21% pharmacists have knowledge about PGx testing).

Among the 1005 respondents included in the study, 99.1% (*n* = 996) were pre-knowns, i.e., respondents who had heard of PGx testing before our survey. A quarter of the pre-knowns had a good grasp of PGx testing, which was higher than results from high-income countries (5.0% in the US and 12.5% in Japan) [24]. This may be explained by our study sample having a general tendency to accept new technologies compared with the general pharmacist population in China, which might have led to an over-estimation of knowledge about PGx testing. Unlike western countries, where pharmacy services are mainly provided in community pharmacies [29], in China, pharmacy services are mainly provided in secondary and tertiary hospitals. Secondary hospitals located in counties or districts mainly provide medical services to local residents. Tertiary hospitals are the highest-level hospitals in China and include national, provincial, municipal, and teaching hospitals [30]. Our respondents all worked at tertiary hospitals, with relatively easy access to high-end healthcare resources and innovative care [31]. They might thus have a more positive attitude toward PGx testing and were more willing to practice it than pharmacists who worked in medical institutions of other tiers (i.e., primary care facilities and secondary hospitals) in China.

The 996 pre-knowns generally had a positive attitude towards PGx testing. Respondents who had a higher level of knowledge about PGx testing were more likely to hold a positive attitude toward it. In Japan [24], surveyed pharmacists were more positive about the significance of PGx testing in improving (95.3%) or attenuating treatment efficacy (91.7%) compared with our respondents. A study in Egypt [32] also showed a higher proportion of respondents (95.2% vs. 59.0%) who agreed that PGx testing would improve pharmacists’ capacity to control expenditures on drug therapies. PGx testing also appeared to be valued differentially in different therapeutic areas. Our respondents assigned the highest value to the use of PGx testing in targeted oncology therapies. Furthermore, clinical pharmacists in oncology department tended to score oncology therapeutic areas higher than therapeutic areas in which they did not specialize, which may suggest that specialty-targeted pharmacogenomic training would be more helpful for clinical pharmacists.

The high cost or lack of insurance coverage of PGx tests, lack of trained professionals, and lack of relevant knowledge were cited as the three most important factors hindering the uptake of PGx testing. In a Japanese study [24] in 2021, “not covered by insurance”, “requiring expenses for analysis”, and “lack of workforce” were also the three main factors that hampered the use of PGx testing in routine clinical practices. In Thailand, most respondents (74.0%) felt extremely or moderately concerned about the reimbursement for PGx testing [25]. To promote the use of PGx testing, China has considered incorporating PGx testing into health insurance schemes. Beijing, as a pilot city, introduced the unified pricing and insurance coverage for PGx testing for tumors (which included pharmacogenomic/germline testing in addition to somatic testing) for the first time in 2019 [33], with the reimbursement rate as high as 90.0%. For cancer patients with their medical insurance registered in Beijing, PGx testing for tumors can be carried out at any hospital in Beijing and reimbursed by their health insurance. On the other hand, ”Guidelines, consensus, and clinical pathways for disease diagnosis and treatment” were the most preferred approaches to learn about PGx testing in this study. In China, there are only six domestic consensus or guidelines [34,35,36,37,38,39], a number much smaller than in the United States and other developed countries. The domestic consensus and guidelines are developed by concerned associations and the National Health and Family Planning Commission, and they cover cardiovascular, oncology, and specific assays as well as provide information on evidence-based and result interpretation of PGx testing for specific genes [4,40,41,42]. Given that our results showed that respondents’ understanding of domestic guidelines and expert consensus was better than their knowledge of the three well-known international standard guidelines or databases (CPIC, PharmGKB, ClinGen), the lack of domestic guidelines might have impacted the uptake of PGx testing. Likewise, Guo and colleagues [13] conducted a survey on PGx use among physicians, pharmacists, and researchers and found that over 50% of pharmacists recognized the importance of PGx testing. However, the pharmacists also perceived that the lack of clinical studies and domestic guidelines were the main barriers to increasing uptake of PGx testing, which is consistent with our findings.

The vast majority of respondents hoped to receive more training in PGx testing in the future, with “Short-term intensive training”, “Online video courses” and “Expert lectures” being the three most popular approaches. In other studies, online courses were also a popular way of training, which may be due to the fact that pharmacists had a busy schedule, and online training and materials allowed for swift scheduling. In comparison, the preferred training modes by pharmacists in Malaysia [43] were undergraduate PGx education (54.0%) and prior PGx education (38.9%). A study in Australia [27] showed that most respondents believed that the best setting to educate pharmacists was at university during a Bachelor of Pharmacy degree (66.7%) and after registration, as workshops and seminars which contributed to pharmacists’ continuing education program (79.0%). However, only 10.7% and 14.4% had studied PGx testing-related courses in undergraduate and graduate programs in our results, respectively. A similar tendency was also found in the US and Japan [24], with 26.1% of the surveyed pharmacists learning about PGx in their undergraduate courses in the US, compared with 12.9% in Japan [24] and 31.6% in Australia [27]. Future educational efforts should consider the above findings to train clinical pharmacists about PGx testing and incorporate PGx into undergraduate and graduate courses.

Our survey has some limitations. Firstly, the respondents in our study were all practicing at tertiary hospitals, who generally had a higher level of knowledge than professionals practicing at health facilities of other tiers. Although most PGx tests were conducted in tertiary hospitals [30], our results may not represent the knowledge of and attitudes toward PGx testing of the broader pharmacist population across China. Secondly, the knowledge of PGx testing was assessed through self-evaluation, which might be vulnerable to social desirability bias [44] and not truly reflect the knowledge level of Chinese clinical pharmacists on PGx testing.

## 5. Conclusions

This study surveyed clinical pharmacists across China and found that most Chinese clinical pharmacists who responded had a positive attitude toward PGx testing. However, they were less familiar with international standard guidelines and databases about PGx testing. The high cost of testing, shortage of relevant professionals, and lack of insurance coverage were perceived as main challenges that hindered uptake of PGx testing in routine clinical practices. Future courses and trainings materials for health professionals should incorporate information about PGx and PGx testing.

## Figures and Tables

**Figure 1 jpm-12-01348-f001:**
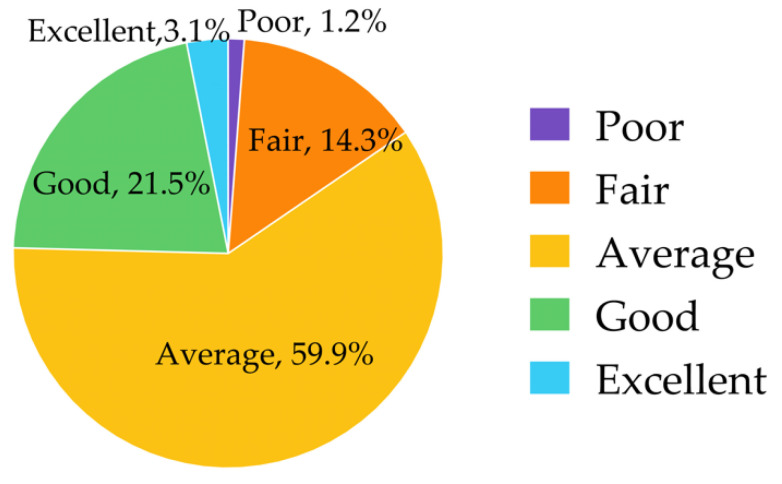
Knowledge of pharmacogenomic testing.

**Figure 2 jpm-12-01348-f002:**
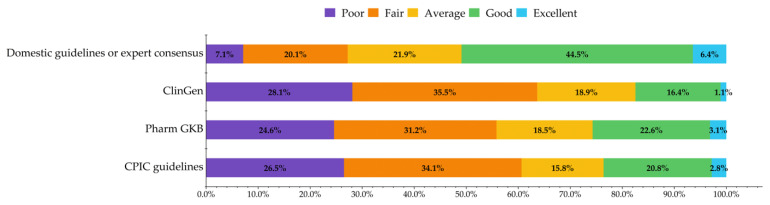
Knowledge about PGx-related guidelines, monographs, and databases.

**Figure 3 jpm-12-01348-f003:**
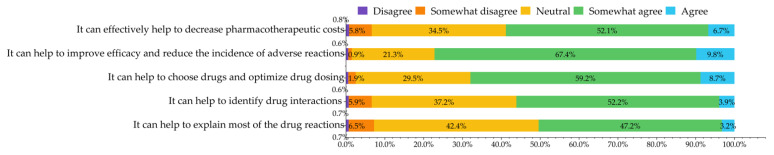
Attitudes towards the role and clinical effect of pharmacogenomic testing.

**Table 1 jpm-12-01348-t001:** Characteristics of study respondents.

		All		Have Heard of Pharmacogenomic Testing Before	
		*n* ^1^	%	*n*	%
Sum		1005	100.0	996	99.1
Gender					
	male	244	24.3	241	24.2
Age					
	≤30	183	18.2	178	17.9
	31–35	472	47.0	468	47.0
	36–40	225	22.4	225	22.6
	>40	125	12.4	125	12.6
Highest qualification					
	bachelor degree or below	332	33.0	327	32.8
	master’s degree	596	59.3	592	59.4
	doctoral degree	77	7.7	77	7.7
Overseas education background ^2^					
	for a degree	7	0.7	7	0.7
	exchange and visit	50	5.0	50	5.0
Years of work experience ^3^					
	≤5 years	212	21.1	208	20.9
	5–10 years	406	40.4	401	40.3
	>10 years	387	38.5	387	38.9
Professional title					
	junior pharmacist or below	152	15.1	144	14.5
	middle level pharmacist	633	63.0	630	63.3
	associate chief or chief clinical pharmacist	220	219	182	18.3
Relationship between practicing hospital and medical college					
	neither affiliated nor teaching hospital	150	14.9	144	14.2
	teaching hospital ^4^	216	21.5	213	21.4
	affiliated hospital ^5^	609	60.6	609	61.1
GDP rank of the practicing province					
	low	265	26.4	261	26.2
	middle	332	33.0	328	32.9
	high	408	40.6	407	40.9

Annotation: ^1^ N refers to the number of respondents. ^2^ Overseas education background refers to whether respondents have studied abroad or participated in an overseas exchange program. ^3^ Years of work experience refers to the duration of working as a pharmacist. ^4^ The affiliated hospitals are part of the medical school and have affiliation with the medical school, including the affiliated general hospitals that are responsible for the whole clinical teaching (theoretical teaching, clinical internship, and graduation internship) and the affiliated specialized hospitals that are responsible for part of the clinical teaching. ^5^ Teaching hospitals are general hospitals or specialized hospitals that have established stable teaching relationships with medical colleges and universities.

**Table 2 jpm-12-01348-t002:** Value of pharmacogenomic testing in different treatment areas (scale 0–100).

Areas	Medium	0–20	20–40	40–60	60–80	80–100
(1) Targeted Oncology Therapy	78.65 ± 22.41	31 (3.1%)	52 (5.2%)	97 (9.7%)	275 (27.6%)	541 (54.3%)
(2) Cardiovascular disease	66.80 ± 23.65	54 (5.4%)	81 (8.1%)	213 (21.4%)	347 (34.8%)	301 (30.2%)
(3) Rheumatic immune diseases	65.51 ± 25.30	69 (6.9%)	107 (10.7%)	215 (21.6%)	303 (30.4%)	302 (30.3%)
(4) Psychiatric and neurologic conditionsneurology	64.51 ± 25.02	75 (7.5%)	110 (11.0%)	212 (21.3%)	315 (31.6%)	284 (28.5%)
(5) Infectious diseases	59.11 ± 26.31	97 (9.7%)	151 (15.2%)	251 (25.2%)	272 (27.3%)	225 (22.6%)
(6) Pain Treatment	58.79 ± 25.47	99 (9.9%)	139 (14.0%)	271 (27.2%)	283 (28.4%)	204 (20.5%)
(7) Other fields	47.83 ± 28.42	216 (21.7%)	166 (16.7%)	254 (25.5%)	231 (23.2%)	129 (13.0%)

**Table 3 jpm-12-01348-t003:** Scores of clinical pharmacists from different departments on the value of pharmacogenomic testing in different therapeutic areas (the most relevant departments to the disease areas were presented).

	Area	Targeted Oncology ^2^	Cardiovascular	Rheumatic Immune	Psychiatry and Neurology	Infectious Diseases	Pain Treatment
Department ^1^							
Card		77.40 ± 24.11	70.84 ± 25.09	63.35 ± 25.90	64.74 ± 22.43	60.54 ± 26.09	60.74 ± 25.90
Cere/Neur		70.99 ± 22.02	63.61 ± 21.23	64.87 ± 24.33	65.77 ± 20.34	57.64 ± 24.30	56.74 ± 22.60
Onco		83.59 ± 21.12	61.75 ± 26.37	60.10 ± 26.38	57.61 ± 26.98	54.75 ± 27.40	55.55 ± 26.98
Psy		73.86 ± 26.24	60.93 ± 29.91	63.57 ± 31.50	72.14 ± 23.70	55.79 ± 33.47	63.36 ± 32.17
Rheu		84.29 ± 14.57	67.14 ± 22.91	73.00 ± 17.15	68.71 ± 23.31	44.00 ± 25.01	61.00 ± 21.32
Infe		83.20 ± 22.67	70.80 ± 24.90	69.25 ± 28.36	70.40 ± 23.95	66.00 ± 25.26	68.00 ± 18.60
Pain		76.38 ± 34.72	64.54 ± 29.57	64.46 ± 31.68	65.54 ± 30.69	57.69 ± 29.05	53.31 ± 34.67

Annotation: ^1^ Cardiovascular (Card), Cerebrovascular/Neurology (Cere/Neur), Oncology (Onco), Psychiatry (Psy), Rheumatology (Rheu), Anti-infectives (Infe). ^2^ The association between respondents’ clinical practice specialties with their scoring of the value of PGx testing in the same areas was tested by Kruskal-Wallis H Test, which showed that the oncology pharmacist–oncology PGx association was the only one that was statistically significant.

## Data Availability

Not applicable.

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
