# Peer review of "Clinical Pharmacists’ Knowledge of and Attitudes toward Pharmacogenomic Testing in China"

_jpm, 2022, doi:10.3390/jpm12081348_

Round 1

Reviewer 1 Report

The authors present a larger scale survey of clinical pharmacist impressions of pharmacogenetic testing amongst clinical pharmacists in China. There has been limited insight into this previously. Please see below for comments/suggestions/revision recommendations.

Major revisions recommended:

- Include additional detail with respect to surveying approach and response. 

- Please include survey instrument.

Overall comments, suggestions, revisions.

Pg 2, line 44-47, this list is somewhat misleading. Would revise to clarify type of clinical information provided by CPIC, DPWG, and other clinical PGx organizations versus the information curated from regulatory agencies – i.e. prescribing recommendations vs. drug label annotations.

Pg 2, line 56: Recommend reviewing and and also referring to the 2022 ASHP statement.

Pg 2, line 59: Please revise to improve clarity of “little turnaround time.” I don’t believe this is what the referenced paper was trying to indicate.

Pg 2, line 65: Please revise last sentence of the paragraph since it’s not entirely true that there is a lack of research of clinical pharmacist’s knowledge of PGx, or at least specify that pharmacist knowledge has been previously evaluated but not at the same scale as this survey. Reference #12 was another survey (albeit not purely of pharmacists but included pharmacists).

Pg 2, line 71-72. Please cite previous studies’ questionnaires that are being referred to.

Pg 2, line 77 – pg 3, line 78. Would be very useful to include the complete survey questions used for this study either in the text or in the supplementary file to better orient the readers.

Pg 3, line 87 – sampling methods. Please include more details of the parameters of the random sampling – how was it determined what % or # of participants was aimed for.

Pg 3, line 100. Please check reference.

Pg 3, line 111: Please re-check and clarify if S3X is Shanxi or ShAAnxi. S1X previously is referred to as Shanxi also.

Pg 3, line 114-115. Please include statistical tests used as applicable.

Pg 4, line 120. Please include # of individuals this survey was sent to as well as the # of those who may have started the survey but didn’t complete it so that the reader can better appreciate the response rate and completion rates in relation to the 1005 completed responses received.

Pg 4, Table 1: Consider including the clinical specialty reported by respondents in this table. This may be useful to show here especially as there is discussion later in the manuscript about the clinical areas that respondents felt more comfortable with applying with PGx. A Under Highest qualification, need to correct “blow” misspelling to “below.” Please consider revising “doctor’s” degree to “doctoral degree” to reduce risk of confusion with a medical degree.

Pg 5, lines 134-136. This introductory sentence should be in the method’s section.

Pg 6, line 144: For p=0.00, consider using P<0.001, NS, or similar wherever very small p-values are used throughout the manuscript.

Pg 7, line 186. The text states “three” questions but in the figure underneath this it shows five statements. Please clarify discrepancy.

Pg 8, line 210-212. Recommend reporting if there was any association with respondents’ clinical practice specialties to see if there was any significant association with their specialty vs their scoring of the value of PGx testing in the areas questioned about. For example, did oncology pharmacists tend to rate oncology therapy higher than the therapeutic areas that they don’t specialize in? This could really add value to the paper in understanding how to approach PGx training/education (if specialty-targeted or generalized training would be more appropriate).

Pg 11, line 281-282. Please cite domestic guidelines if possible. If not citable, please describe more what the therapeutic/topic areas of domestic guidelines entail, how developed, what information or type of recommendations are provided.

Pg 12, line 328-329. Please clarify if PGx tumor testing also includes pharmacogenetic/germline testing in addition to somatic testing.

Supplementary materials: As mentioned above, please include the survey instrument. For Figure S1, it is unclear why the order is the way it is (not organized by # of respondents – i.e. ZJ and Gx are not ordered correctly; or by department; Cards should be after Onco and Resp?) – please resort.

Author Response

Dear reviewer:

We appreciate your kind consideration and thorough review of this article. Your comments are very instructive and helpful. We are very sorry for the errors in the paper. We have made comprehensive and detailed revisions according to your suggestions. Please see the attachment.

Reviewer 2 Report

Authors describe their study on assessing the knowledge and attitudes of clinical pharmacists on pharmacogenomic testing. I have the following comments:

1. English can be improved. There are long sentences in many instances in the manuscript that can be shortened for easy readability. 

2. It would be good to include a copy of the survey is journal allows that. 

3. Why did you decide to report on results of only the first five parts?

4. Methods section can be shortened. Clarify what you mean by "RMB10"?

5. Avoid repeating results in the discussion. 

6. Avoid repeating information in table/figure and text part of the results. 

7. Table 1 - clarify what you mean by "N" as foot note. Spell check "blow".

8. Clarify what you mean by "overseas education background".

9. Years of work experience  - this include only as pharmacist or any other work?

10. "Relation between prac..." - need to clarify what this means. 

11. I am not sure what you mean by "new technologies" in the survey. what are you referring to ? how would have the respondents perceived that? how are you relating that to PGX?

12. Figures 2 and 3 need to be clear and explained better. 

13. Result section need to be focused and short as well. Currently it is easy for readers to get lost. 

14. Discussion is long. Please shorten the discussion and discuss relevant aspects. 

Author Response

Dear reviewer:

We appreciate your kind consideration and thorough review of this article. Your comments are very instructive and helpful. We have, therefore, revised the manuscript accordingly. Please see the attachment.

Reviewer 3 Report

Well written work and the topic is very interesting. 

Author Response

Dear reviewer:

Thanks very much for taking your time to review this manuscript.

Round 2

Reviewer 1 Report

Dear authors, thank you for your responses to the comments and for including clarifications, the survey, survey response, and including citations/references which can provide further resources to review for particularly interested readers. My feedback at this point are mostly minor and related to grammar and content placement.

1. Overall, would recommend reviewing for minor grammar edits throughout. Some examples include

- Pharmacogenomic testing (as originally used in the original submission is more commonly used as compared to “pharmacogenomics” testing.

-      -"Pharmacogenomics” used as a standalone noun referring to the study/field of pharmacogenomics is correct.

-      -"Pharmacogenomics research” is fine.

-      -Check verb tenses throughout for consistency.

2. Page 2, lines 189-194. Thank you for including survey response rate information. Two issues to address:

-      - The survey was clicked (assuming this means opened) by 1560 people, but was this the total # of individuals the survey was sent to or was it more? Would recommend also including this # beyond how many people opened/clicked the survey link if able to, although from the Appendix it looks like a general survey link was used instead of generating individual unique survey links. If a general link was used, then the 64.4% would probably be better described as “survey completion rate” rather than “response” rate.

-      -Since the information provided on page 2 is mainly results data on survey response/completion/etc, this content should be  moved to the Section 3 results section and combined with the duplicative survey response rate information on page 3 lines 468-469. Can consider separating the survey response content into a separate 3.x section (example “survey response rates and inclusion” or similar) before the existing 3.1 respondent characteristics section. On page 2 in section 2.2, would recommend briefly state the # of people the survey was sent to in order to set the stage for sharing survey response data in section 3 (results section).

3. Page 5 lines 755-756 – page 6 lines 951-953. Recommend explicitly stating that the oncology pharmacist – oncology PGx association was the only one that was statistically significant. That or indicate in table 3.

4. I overall appreciate the discussion section. It may be a bit long for some, but I think it provides insight into attitudes towards and context of PGx testing and clinical resources in China, particularly the newly inserted commentary.

Author Response

Dear reviewer,

Thank you for your comments concerning our manuscript entitled “Clinical pharmacists’ Knowledge of and Attitudes towards Pharmacogenomic Testing in China”(ID: JPM-1802820). Those comments are all valuable and very helpful for revising and improving our paper, as well as the important guiding significance to our research. We have studied the comments carefully and have made corrections which we hope to meet with approval. Revised portions are marked in red in the paper. Please see the attachment.

Reviewer 2 Report

They have responded appropriately

Author Response

Dear reviewer,

Thanks for your positive comments.